# An Eco-Friendly and Innovative Approach in Building Engineering: The Production of Cement–Glass Composite Bricks with Recycled Polymeric Reinforcements

**DOI:** 10.3390/ma17030704

**Published:** 2024-02-01

**Authors:** Marcin Małek, Janusz Kluczyński, Katarzyna Jasik, Emil Kardaszuk, Ireneusz Szachogłuchowicz, Jakub Łuszczek, Janusz Torzewski, Krzysztof Grzelak, Ireneusz Ewiak

**Affiliations:** 1Research Laboratory, Faculty of Civil Engineering and Geodesy, Military University of Technology, Gen. S. Kaliskiego 2, 00-908 Warsaw, Poland; marcin.malek@wat.edu.pl (M.M.); emil.kardaszuk@wat.edu.pl (E.K.); 2Institute of Robots & Machine Design, Faculty of Mechanical Engineering, Military University of Technology, Gen. S. Kaliskiego 2, 00-908 Warsaw, Poland; katarzyna.jasik@student.wat.edu.pl (K.J.); ireneusz.szachogluchowicz@wat.edu.pl (I.S.); jakub.luszczek@wat.edu.pl (J.Ł.); janusz.torzewski@wat.edu.pl (J.T.); krzysztof.grzelak@wat.edu.pl (K.G.); 3Department of Imagery Intelligence, Faculty of Civil Engineering and Geodesy, Military University of Technology, Gen. S. Kaliskiego 2, 00-908 Warsaw, Poland; ireneusz.ewiak@wat.edu.pl

**Keywords:** additive manufacturing, cement–glass composite bricks, waste disposal, PET-G, fused filament fabrication, digital image correlation

## Abstract

Cementitious–glass composite bricks (CGCBs) with 3D-printed reinforcement structures made of PET-G could be an innovative production method that relies on recycling glass waste (78%) and PET-G (8%). These bricks offer a promising solution for the construction industry, which has a significant impact on climate change due to its greenhouse gas emissions and extensive use of natural aggregates. The approach presented in this article serves as an alternative to using conventional building materials that are not only costlier but also less environmentally friendly. The conducted research included mechanical tests using digital image correlation (DIC), utilized for measuring deformations in specimens subjected to three-point bending and compression tests, as well as thermal investigations covering measurements of their thermal conductivity, thermal diffusivity, and specific heat. The results highlighted the superior thermal properties of the CGCBs with PET-G reinforcements compared to traditional cementitious–glass mortar (CGM). The CGCBs exhibited a 12% lower thermal conductivity and a 17% lower specific heat. Additionally, the use of specially designed reinforcement substantially enhanced the mechanical properties of the bricks. There was a remarkable 72% increase in flexural strength in the vertical direction and a 32% increase in the horizontal direction.

## 1. Introduction

Additive manufacturing (AM), commonly known as 3D printing, is gaining substantial importance in the construction industry, bringing about a revolution in traditional design and construction methods. Among AM technologies, the Material Extrusion (MEX) process and, more specifically, the fused filament fabrication (FFF) technique are extensively employed, owing to the numerous advantages they provide [1,2,3]. AM technologies stand out for their cost-effectiveness and wide range of available materials. They enable the creation of three-dimensional structures layer by layer, a crucial aspect in reshaping how materials and building forms are developed. Given the challenges of efficiency, sustainable development, and structural flexibility, there is growing interest in new solutions. AM provides designers and engineers with the tools to craft more efficient and sustainable structures while cutting down on the production time and material usage.

In construction, 3D printing is used to create reinforcements, addressing the low plasticity of cementitious materials. This method serves as an alternative to traditional steel rods, eliminating the risk of corrosion. Another remarkable benefit of AM in construction is waste reduction. Traditional construction generates substantial and costly waste, which is environmentally unfriendly. AM’s ability to produce the exact amount of needed material significantly contributes to reducing construction waste and supports sustainable resource management through material reuse. In the FFF technique, polymers are the primary materials employed. The use of recycled polymers involves obtaining granules or powder, which undergo processing to form filaments specifically designed for application in the FFF technique for AM processes. Woern et al. [4] conducted a comparison between the use of the original polylactide (PLA) and 3D printing using four recycled polymers, which included two widely used printing materials (PLA and acrylonitrile-butadiene-styrene (ABS)), along with two common plastic wastes (polyethylene terephthalate (PET) and polypropylene (PP)). The tests indicated that a diverse range of recycled materials can be employed for printing without compromising the mechanical properties of the prints. In related studies, the authors of [5] demonstrated a lower elongation at break for recycled Polyethylene Terephthalate Glycol (PET-G) material compared to its original form. Equally, the authors of [6,7] noted that this material can undergo recycling without a decline in its mechanical properties. Additionally, PET-G exhibits low shrinkage during the additive manufacturing process, does not absorb moisture, and demonstrates a good creep resistance under constant loading conditions [7]. Other studies have demonstrated that, in the case of composite bricks, the primary factor influencing their strength is the scaffolding design rather than the material used for production [8]. In their research, the authors of [9] examined two topology configurations—cubic and three-dimensional octets—made from two materials: ABS and PLA. The study also considered the impact on the mechanical properties, revealing that the final mechanical properties were primarily affected by the scaffolding topology, not the material used for printing the scaffolding. Qin et al. [5] delved into various structures of polymer reinforcements based on basic shapes such as squares, diamonds, and hexagons. Depending on the type of scaffolding used, the authors obtained different mechanical properties for the final bricks. PET-G serves not only as reinforcement but also as a substitute for aggregates [10,11], reducing the reliance on natural aggregates. This not only benefits the natural environment but can also enhance the bricks’ thermal and mechanical properties [12,13]. In a different study [14], it was demonstrated that the addition of PET-G fibers to concrete significantly improves its resistance to impact loads and its ability to absorb energy under low-velocity impact conditions.

In further research [11], fresh concrete containing PET-G was found to have a lower density. Concrete with the addition of PET particles exhibited a lower modulus of elasticity and tensile strength compared to its conventional counterparts. Although there was an initial tendency for the compressive and flexural strength to increase, this trend diminished over time. In another study, the authors [15] replaced 50% of the sand with PET-G granules with a maximum particle size of 5 mm in diameter. This substitution did not affect the compressive strength or flexural strength of the resulting composites, indicating that plastic bottles made of PET, when divided into small particles, can be an interesting and cost-effective solution with consistent properties in the production of cement bricks. An alternative approach involves using a glass cullet as an aggregate. Previous analyses of incorporating waste glass as aggregates or cement components into standard concrete mixes suggest its effectiveness in glass recycling. Laboratory studies have indicated that the properties of concrete with glass as an aggregate primarily depend on the particle size and the amount of glass in the concrete mixes. However, the use of coarser glass aggregates in concrete can lead to an alkali–silica reaction (ASR), thereby reducing the strength and durability of the concrete [16].

The utilization of recycled glass in dry concrete mixes has not been as extensively studied as in conventional concrete. When used as an aggregate in concrete blocks, glass results in reduced water absorption and improved wear resistance due to the high hardness of the glass components. Turgut et al. [16] explored various levels (10%, 20%, and 30%) of fine glass (<4.75 mm) and coarse glass (4.75–12.5 mm) as a substitute aggregate in the production of concrete blocks. Their findings demonstrated that the maximum compressive strength was achieved with 20% fine glass while increasing the content to 30% slightly reduced the strength. Conversely, increasing the content of coarse glass from 10% to 30% gradually increased the compressive strength of the paving blocks.

The construction industry is a key player in the realms of climate change and recycling, emitting significant greenhouse gases and consuming vast quantities of natural aggregates. This research addresses these challenges by proposing an innovative and cost-effective solution—CGCBs with an internal scaffolding made of recycled PET-G. This construction material, primarily composed of waste glass and processed PET-G scaffolding, has the potential to bring about substantial improvements in addressing current challenges and enhancing the mechanical properties of bricks.

Previous studies by the authors [17,18] introduced a method for creating composite cement–glass bricks, which was further modified in this study to enhance their stability, mechanical properties, and durability. The research offers an alternative to traditional building materials by introducing CGCBs with internal scaffolding based on recycled PET-G from additive manufacturing (AM). The new material, comprising predominantly secondary materials (75% glass, 8% PET-G), presents a potentially more economical and sustainable option compared to the traditional materials. The aim of the study is to expand the knowledge base in the research domain encompassing the current challenges in waste utilization. Despite the increasing interest in the use of aggregates and fibers derived from waste in concrete research, there remains insufficient knowledge regarding other potential areas of waste disposal. Specifically, there is a lack of research focused on 3D-printing technology, primarily concentrating on scaffold printing. The presented research addresses these knowledge gaps by providing results related to the material, thermal, and mechanical properties of composite cement–glass bricks with 3D-printed PET-G scaffolding, predominantly constructed from secondary materials. The conducted research included mechanical tests using digital image correlation (DIC), utilized for measuring the deformations in specimens subjected to three-point bending and compression tests, as well as thermal investigations, covering measurements of their thermal conductivity, thermal diffusivity, and specific heat.

## 2. Materials and Methods

### 2.1. Characteristics of the AM Process

The polymer scaffolds were produced using the FFF method, which was previously designed in the Netfabb software. (Version Premium 2024) The material used to produce the samples was PET-G in the form of 1.75 mm diameter filaments (Spectrum Filaments Ltd., Pęcice, Poland). The parts were produced using a Prusa Original MK3s printer (Prusa Research, Prague, the Czech Republic) and using the following parameters:Nozzle temperature: 240 °C,Build plate temperature: 85 °C,Nozzle diameter: 0.4 mm,Layer thickness: 0.2 mm,Part cooling intensity: 40%,Printing speed: 50 mm/s,Infill density: 100%.

For each type of test, five samples were printed so that the tests conducted would be reliable. Figure 1 shows the fabricated structure.

### 2.2. Cement–Glass Mortar Filler

The filler used for the printed scaffolds was a cement–glass mortar based on Portland cement (CEM I 42.5R NA [19]). Its specification was determined according to the applicable standards [20,21]. It comprised tap water, fine glass powder waste (particle size below 0.1 mm), and waste glass aggregate (particle size ranging from 0.1 mm to 0.2 mm). To demonstrate the appropriate aggregate matching, a curve was established for the glass sand with defined lower and upper bounds [22,23], as shown in Figure 2.

With a maximum size of 2 mm for the glass cullet, the curve aligns within this range, indicating a compacted aggregate stack. The glass cullet used in the research exhibited an uneven surface shape due to either a mechanical or implosive crushing method. Brown, green, and transparent glass granules were also utilized in the study. Additionally, a third-generation liquid admixture based on modified polymers was used to maintain a low water-to-cement ratio (*w*/*c* = 0.29) and substantially reduce the amount of water in the cementitious–silica mortar. Table 1, Table 2 and Table 3 contain the chemical compositions and physical properties of the cement and glass waste. XRF was used to determine the elemental composition of the cement sample. The XRF setup includes an X-ray tube that generates the primary beam (the production system), equipped with primary collimators, crystals, secondary collimators, and detectors. The X-rays generated by this system excite the atoms in the sample, leading them to emit radiation as they return to their stable state. An XRD system analyzes this emitted radiation. The XRF measurements were conducted using the ARL 9900 (Thermo Fisher, Waltham, MA, USA), employing the monochromatic radiation K α1 of cobalt (wavelength = 1.788996 Ǻ).

### 2.3. Description and Manufacturing Process of CGM

The formulation of the cementitious–silica mortar was developed using well-established techniques employed in crafting high-quality composites [25]. A design approach that combines computational and experimental methods was utilized. Currently, numerical methods cannot guarantee the ability to produce composites with both high strength and consistent results without real-world trials. Therefore, manual extreme condition tests were carried out. The design process incorporated initial calculations and assumptions, which were later refined based on empirical validation using laboratory investigations. The final composition of the CGM mix, used as filler for 3D-printed concrete scaffold bricks, is shown in Table 3. The development of the methodology for designing the composite was carried out using the Bukowski method. Initially, an analysis of the concrete components such as the aggregates, waste, and additional admixtures was conducted. Based on this, a preliminary formula was prepared, assuming the concrete’s exposure classes, planned strength, and the water-to-cement ratio. After creating the samples, they were subjected to tests, after which the formula’s composition was optimized to achieve even higher strength values by adding more superplasticizers and re-sealing the concrete structure. This led to repeating the sample preparation stage and adjusting their composition so that the resulting concrete mass would have the best properties. Following this process, the spatial structure was cast, and the final samples for testing were obtained. This is a systematic approach that emphasizes a thorough analysis of the components and conditions in which the concrete will be used, allowing for the production of a high-quality and resistant material. This method is particularly useful in designing special concrete or in conditions where the standard approaches do not offer a sufficient performance or durability.

To achieve a uniform mass from all the aforementioned dry components, a high-speed planetary mixer with three agitator speed ranges was utilized, completing the operation within a minute. Upon the introduction of the wet constituents, the mixing process was extended by an additional 4 min, opting for a medium speed to thoroughly blend the components (midway among the three available speeds). Subsequently, the CGM was compacted on a vibrating table, fitting it into molds with pre-printed scaffolds. The top layer underwent vibrational compaction for approximately 30 s. Following this, using a water-moistened regular knife, the top layer of the sample was leveled to the edge of the mold after filling. To mitigate the excessive loss of mixing water and counteract the shrinkage caused by heat during the cement hydration process, absorbent mats were placed over the upper layer of the samples 24 h after their production. A preliminary treatment phase lasting for 12 h was implemented. Following that, the samples were removed from the mold and immersed in water, adhering to the EN 12390-2:2019-07 standard [26]. Consistent laboratory conditions were upheld at a temperature of 21 °C and a humidity level of 50% throughout the entire production process.

### 2.4. Testing of the Mechanical Properties Using Digital Image Correlation

The digital image correlation technique was employed to quantify the deformations in the specimens subjected to both three-point bending and compression tests. These evaluations were conducted using an Instron 8802 testing apparatus (Instron, Norwood, MA, USA). In the three-point bending tests, a configuration was utilized with support points spaced 120 mm apart. The strain field for the three-point bending test was analyzed on specimens measuring 40 mm × 40 mm × 160 mm. For the compression tests, specimens with dimensions of 40 mm × 40 mm × 40 mm were employed. To prepare the samples for digital image correlation (DIC) measurement and examination, a layer of white flexible paint was applied to their surface, which was then marked with discernible black dots. The scientific investigation involving digital image correlation was conducted using advanced equipment from Dantec Dynamics (Dantec, Ulm, Germany)–shown in Figure 3, along with ISTRA 4D software (Version 4.4.1x86). This software facilitated visual representation of the fracture progression within the samples through insightful analysis of the deformation fields.

### 2.5. Research on the Physical Properties of Composite Cement–Glass Bricks

In the conducted research, the thermal conductivity, diffusivity, and specific heat of the cementitious–glass material (CGM) samples were accurately determined to characterize their thermal properties. An ISOMET 2114 analyzer from Applied Precision Ltd., Bratislava, Slovakia, was employed for measuring these parameters. The analyzer, equipped with a resistive heater, facilitated the precise determination of the material’s temperature response to the heat flow impulses through the sample. Additionally, a 60 mm diameter probe analyzer was utilized for the measurements.

A total of 10 cubes, each with dimensions of 150 mm × 150 mm × 150 mm, were tested, allowing for a comprehensive assessment of all the thermal properties of the cementitious –glass mortar. Density measurements of the cured samples were also carried out, with the density calculated as the ratio of the mass to the sample volume. For accurate measurements, VIBRA scales from Krakow, Poland, and electronic calipers (TOYA, Wroclaw, Poland) were used.

## 3. Results and Discussion

### 3.1. Thermal Properties of the Cured CGM and CGCBs

The tested samples were examined in terms of their thermal properties. Ten measurements were conducted to determine the thermal properties of the cured samples. The obtained results are presented in Table 4.

The average thermal conductivity of the final CGCBs was 0.87 ± 0.05 W/mK, indicating a 12% lower value compared to the cement–glass mortar. The thermal diffusivity was 0.64 ± 0.03 µm^2^/s, showing a less than 5% increase compared to the cement–glass mortar. In terms of specific heat, the value was 1.36 ± 0.01 MJ/m^3^K, representing a decrease of about 17% compared to the mortar. It is worth noting that the samples tested by Małek in [22] exhibited inferior thermal properties, while the samples tested by the authors in [23] had the same thermal conductivity value, a lower thermal diffusivity (8%), and a higher specific heat (4%).

### 3.2. Density

The average density of the CGM samples was 2157 kg/m^3^, categorizing it as standard concrete. In contrast, the CGCB samples had a lower density of 1982 kg/m^3^, classifying it as lightweight concrete D2.0. The incorporation of scaffolding into the sample reduced its density compared to the CGM by 175 kg/m^3^ (1.14 times). This reduction stems from the limited amount of plastic material used compared to in the traditional cement–glass mortar. To accommodate the 3D-printed scaffolding within the brick structure, a portion of the traditional mortar was removed. Air composition analysis revealed that traditional mortar tends to accumulate a greater number of air voids compared to modern mortar, explaining the differences in the density of the cured samples. In prior research, Małek [18,27] presented similar densities for concrete mixtures with glass as a substitute for natural aggregates. A 100% substitution resulted in values of 2050 kg/m^3^ and 2051 kg/m^3^, also corresponding to the standard concrete class.

### 3.3. Strength Parameters with Digital Image Correlation

#### 3.3.1. Three-Point Bending

A three-point bending study was carried out on the manufactured bricks, considering two orientations: horizontal and vertical. The investigation involved concrete samples without reinforcement, polymer reinforcement, and composite bricks with polymer reinforcement. Figure 4 illustrates the obtained results, offering a clearer understanding of the impact of the utilized reinforcement. The curves for the horizontal CGCB samples, with and without reinforcement, show significant differences, indicating the notably superior strength of the designed internal reinforcements compared to those observed in our previous research [17,18] and the reference samples.

In the vertical direction, the CGM exhibited a maximum bending force of 2.5 kN, and in the horizontal direction, it was 2.1 kN. The maximum force in the vertical direction was recorded at a beam deflection of 0.8 mm, over three times greater compared to the value obtained in the horizontal direction, where the maximum force occurred at a deflection of 0.25 mm. The CGCB samples achieved a lower maximum force compared to the reference samples, measuring 2.2 kN for the vertical configuration (a decrease of 12%) and 1.5 kN for the horizontal configuration (a decrease of 28%), with beam deflections at 0.75 mm and 0.25 mm, respectively. Mesh or cellular materials exhibit different cracking mechanisms depending on their internal structure. The shape, size, and distribution of the cells influence their mechanical properties. The digital image correlation (DIC) method allowed us to observe the crack initiation process, forming the basis for a proper interpretation of the cracking process from initiation to complete failure. The DIC analysis for the samples was conducted in the main deformation direction. The scale applied in the images illustrates the range of deformations in the tested samples. Deformation field images were also captured using the DIC technique. The results are presented in Figure 5. The CGCB bricks tested horizontally behaved similarly to the CGM bricks. Cracks occurred in the middle of the samples, and the crack line was curved in both cases. For the CGCB samples in the vertical direction, three crack lines appeared, which were also curved and occurred in the middle of the tested sample. This could be due to a higher number of micro air bubbles in the direction observed during the testing of the CGCB sample in the vertical direction. However, this did not affect the ultimate bending strength. The flexural strength in the horizontal direction was 6.23 MPa, whereas in the vertical direction, it was 8.12 MPa. Compared to the reference sample, this represents an increase of 32% in the horizontal direction and 72% in the vertical direction. Comparing previous studies by the authors in [18], where a structure of polymer reinforcement generated by the program was used, the manufactured bricks also exhibited a lower strength in the vertical direction by 16.9% and in the horizontal direction by 5.92%. In other studies [28], the cement mortar itself achieved a strength value 22.9% higher, while the final bricks with incorporated reinforcements obtained a similar value to the horizontally tested samples and a significantly lower value than the vertically tested sample, by 22%. In Table 5, the obtained flexural strength results are presented, along with the results obtained in other studies.

The investigated composite glass bricks demonstrated a significantly higher flexural strength compared to the glass–cement concrete, irrespective of the sample orientation (Figure 6a,b). However, the maximum force recorded was lower. This reduction in force during the strength tests, regardless of direction, may be attributed to the symmetrical structure of the printed reinforcement, which was identical in both directions. This suggests a proper distribution of the glass–cement mixture throughout the brick and adequate compaction of the elements in both directions. The decline in force also indicates that the glass particles adhere well to the polymer reinforcement, as evident in Figure 6b.

#### 3.3.2. Compressive Strength

For the same combination as in the three-point bending tests, compression tests were conducted. Like in the bending tests, the test results varied. The characteristic trends for each combination are presented in Figure 7. The highest compressive strength was exhibited by the horizontally tested concrete sample, with a value of 53 kN. In contrast, for the vertical orientation, it was significantly lower at 22 kN and was maintained until a deformation of approximately 8–9 mm. The final composite material with reinforcement showed compressive strengths of 43 kN and 45 kN in the vertical and horizontal orientations, respectively. This represented a decrease of 18% compared to the horizontal concrete sample and an increase of over 50% compared to the vertical concrete sample.

The DIC analysis for the compressed samples demonstrated a positive influence on the applied scaffolding construction. Both in the vertical and horizontal orientations, the CGCB samples exhibited an increased deformation area, and the resulting cracks indicated the higher isotropy of the manufactured final bricks compared to the bricks without scaffolding. The cracks in the samples, both before failure and after a 1 mm deformation, displayed a very similar pattern along the entire length of the tested samples in both the horizontal and vertical directions. The results are presented in Figure 8. Comparing the results with previous studies by the authors [17,18], where the cracks in the CGCB bricks were not distributed so evenly, it can be concluded that samples with the designed scaffolding have a more regular and homogeneous structure, which also impacts their final strength.

## 4. Conclusions

This study aimed to evaluate the mechanical and thermal properties of the proposed CGCBs incorporating PET-G printed scaffolding. The results indicated that the CGCBs exhibited superior thermal and mechanical properties compared to the traditional cement–glass mortar and the structures presented in other studies. The addition of glass particles and a PET-G lattice structure enhanced the performance properties of the produced sample bricks.

The conclusions drawn from the conducted research are as follows:PET-G scaffolding in the cement–glass composite bricks reduced their density by 8%, from 2157 kg/m^3^ to 1982 kg/m^3^, compared to bricks without scaffolding.The CGCBs with PET-G printed scaffolding demonstrated improved thermal properties compared to the CGM, with a 12% decrease in thermal conductivity, a 5% higher specific heat, and a 17% reduction in thermal diffusivity.The bricks with the PET-G scaffolding exhibited high strength, with values ranging from 6.23 to 8.12 MPa depending on the direction of testing. The maximum strength was observed at displacements of 0.25 mm and 0.75 mm. The addition of the PET-G scaffolding increased the bending strength by 32% vertically and 72% horizontally.The designed PET-G structure introduced isotropic mechanical properties into the bricks, regardless of the direction of testing and sample deformation.

There is potential to utilize recycled materials, exemplified by PET-G, in the production of composite cement–glass bricks, effectively replacing natural aggregates. This contributes to waste reduction and sustainable development in the construction industry. There are potential possibilities to develop the suggested technology in this research based on topological optimization of the scaffolding structure to fit this solution to certain exact applications in building engineering.

## Figures and Tables

**Figure 1 materials-17-00704-f001:**
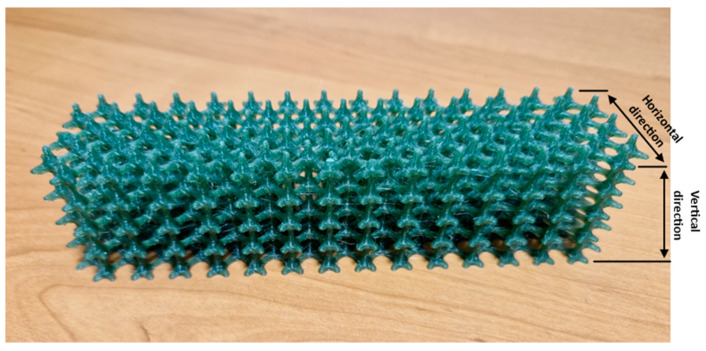
Box lattice mesh structure made of PET-G after the additive manufacturing process.

**Figure 2 materials-17-00704-f002:**
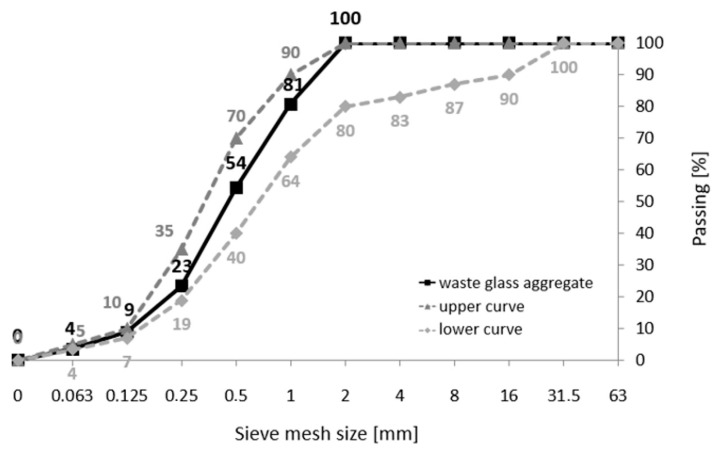
Gradation curve of waste glass aggregate.

**Figure 3 materials-17-00704-f003:**
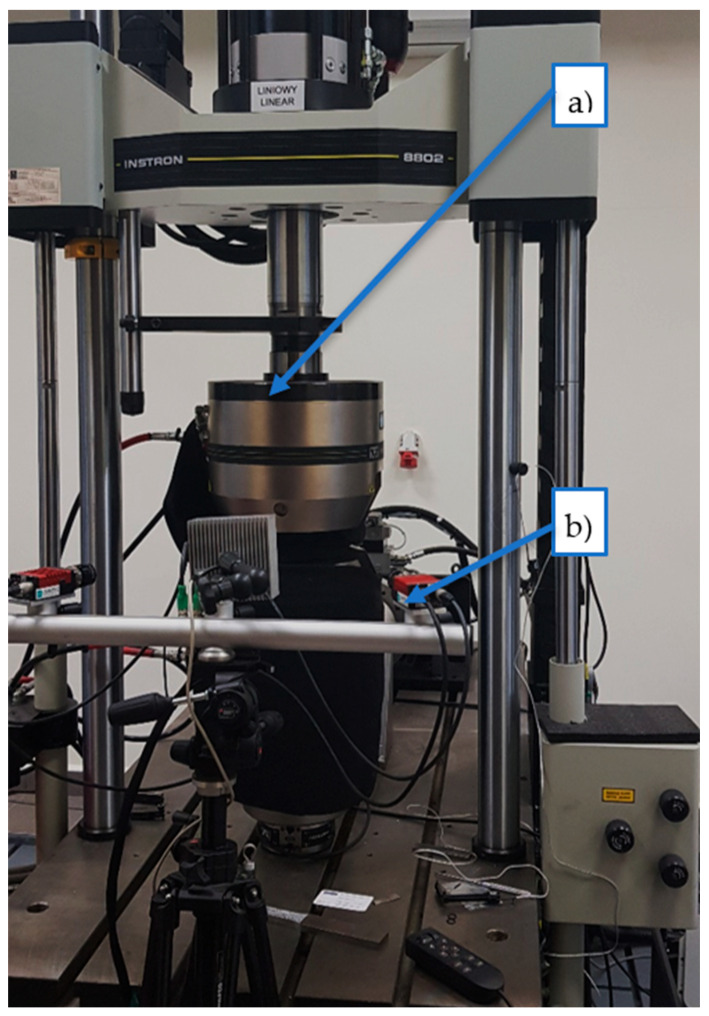
Servo-hydraulic pulsator Instron 8802 for tensile tests (**a**) with digital image correlation system DIC (**b**).

**Figure 4 materials-17-00704-f004:**
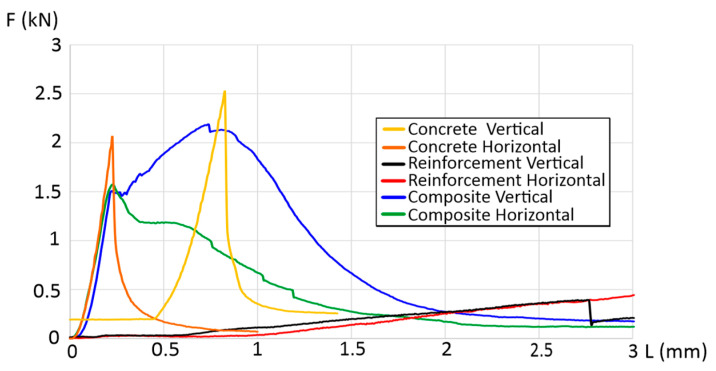
Force dependence as a function of strain in a three-point bending test.

**Figure 5 materials-17-00704-f005:**
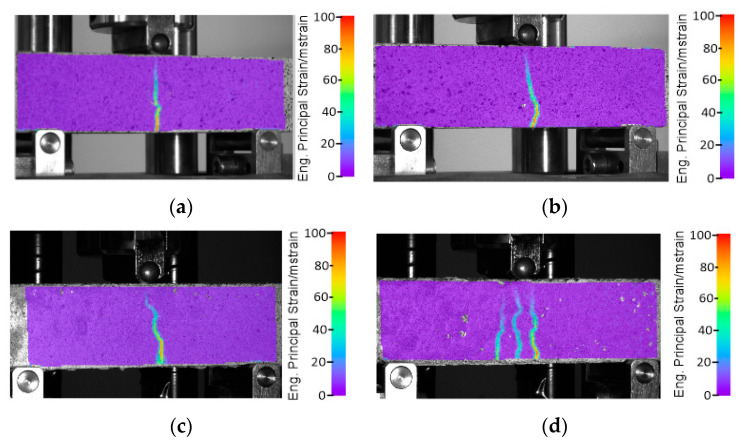
Results of DIC analysis of composite specimens: without (**a**) horizontal or (**b**) vertical reinforcement and with (**c**) horizontal or (**d**) vertical reinforcement.

**Figure 6 materials-17-00704-f006:**
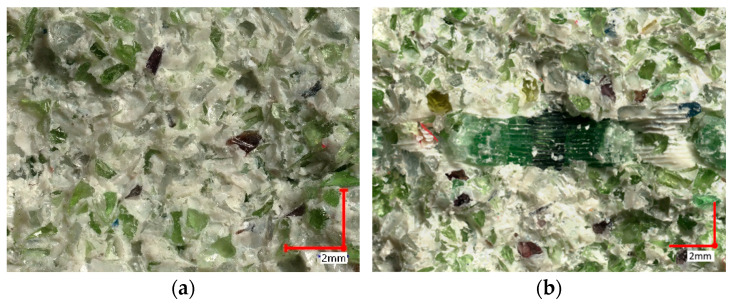
Fracture surface of specimens (**a**) made of cement–glass concrete and (**b**) made of composite tendons with polymer scaffolding.

**Figure 7 materials-17-00704-f007:**
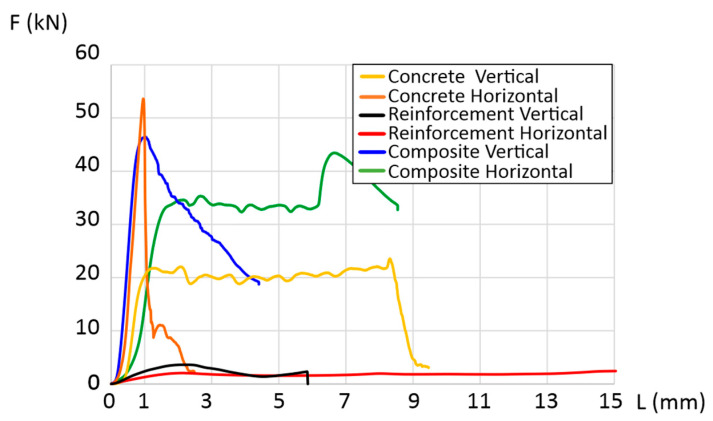
Force dependence as a function of strain in compression tests.

**Figure 8 materials-17-00704-f008:**
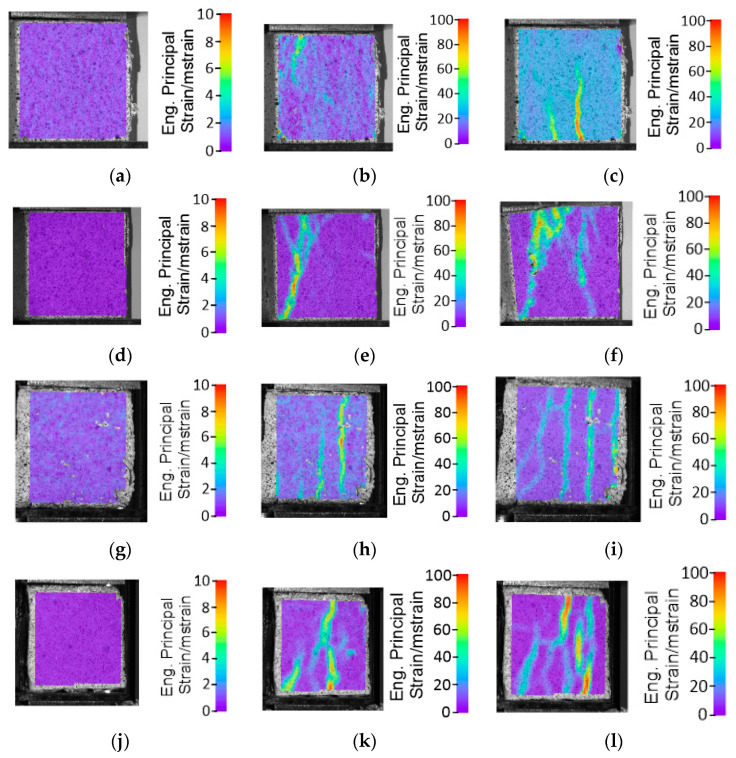
DIC images from the compression strength tests: glass–cement concrete sample in horizontal arrangement (**a**) reference (**b**) after 1 mm deformation (**c**) before failure; glass–cement concrete sample in vertical arrangement (**d**) reference (**e**) after 1 mm deformation (**f**) before failure; composite samples with scaffolding in horizontal arrangement (**g**) reference (**h**) after 1 mm deformation (**i**) after failure; composite samples with scaffolding in vertical arrangement (**j**) reference (**k**) after 1 mm deformation (**l**) after failure.

**Table 1 materials-17-00704-t001:** The chemical composition of cementitious aggregate and glass binder [24].

Compositions		SiO_2_	Al_2_O_3_	Fe_2_O_3_	CaO	MgO	SO_3_	Na_2_O	K_2_O	TiO_2_	Cl
Unit (vol. %)	Cement	19.5	4.9	2.9	63.3	1.3	2.8	0.1	0.9	-	0.05
	Glass	70.0–74.0	0.5–2.0	0.0–0.1	7.0–11.0	3.0–5.0	–	6.0–8.0	7.0–9.0	0.0–0.1	-

**Table 2 materials-17-00704-t002:** The physical properties of cementitious aggregate and glass binder [24].

Property	Specific Gravity [kg/m^3^]	Specific Surface Area [m^2^/kg]	Initial Setting Time [min]	Average Compressive Strength after 28 Days [MPa]
Cement	3090–3190	437	176	68.2
Glass	2450	100	-	-

**Table 3 materials-17-00704-t003:** Proportions of CGM mixture (1 m^3^).

Mix Symbol	Cement	Water	Waste Glass Aggregate [kg]	Chemical Admixture [kg]	Waste Glass Powder
CGM	480	140	1782.2	4.8	117.8

**Table 4 materials-17-00704-t004:** Thermal properties of materials.

Sample Symbol	Thermal Conductivity [W/mK]	Thermal Diffusivity [µm^2^/s]	Specific Heat [MJ/m^3^K]
CGM reference sample	0.99 ± 0.05	0.61 ± 0.03	1.64 ± 0.01
Sample in this study	0.87 ± 0.05	0.64 ± 0.03	1.36 ± 0.01
CGCBs made by Małek et al. in [17]	0.91 ± 0.05	0.66 ± 0.03	1.48 ± 0.01
CGCBs by the authors in [18]	0.87 ± 0.05	0.69 ± 0.03	1.31 ± 0.01

**Table 5 materials-17-00704-t005:** Flexural strength results.

Type of Sample Tested	Average Bending Strength [MPa]	Standard Deviation [MPa]
Reference sample in the study	4.72	±0.13
CGCB vertical sample in this study	8.12	±0.17
CGCB horizontal sample in this study	6.23	±0.16
CGCB vertical sample in the study [18]	6.75	±0.15
CGCB horizontal sample in the study [18]	5.90	±0.14
Reference sample without scaffolding in the study [28]	5.8	±0.15
CGBC sample with reinforcement in the test [28]	6.33	±0.16

## Data Availability

Data are contained within the article.

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
