# Peer review of "An Eco-Friendly and Innovative Approach in Building Engineering: The Production of Cement–Glass Composite Bricks with Recycled Polymeric Reinforcements"

_materials, 2024, doi:10.3390/ma17030704_

Round 1

Reviewer 1 Report

Comments and Suggestions for Authors

Manuscript number: materials-2843989

Manuscript title: Eco-friendly and innovative approach in building engineering: production of cement-glass composite bricks with recycled polymeric reinforcement

This investigation focusses on cementitious-glass composite bricks (CGCB) with 3D-printed reinforcement structures made of PET-G, which offers an innovative production method that relies on recycling glass waste (78%) and PET-G (8%). The overall structure is reasonable, the data are authentic and reliable, and the subject is within the scope of the journal. However, the subtitle is repetitive and the capitalisation should be consistent. Detailed comments are shown as follows:

(1) Initial capitalisation in titles should be consistent.

(2) The methodology and purpose of the study in the abstract were not detailed enough.

(3) Line 117, and Line 136, "2.1 ***" was repeated.

(4) Line 212, "3. RESULTS AND DISCUSSION" was revised to "3. Results and discussion".

(5) Line 289, "Fig. 4. Results of DIC analysis ......", DIC needs to be written as Digital Image Correlation to make the figure title self-explanatory. 

(6) Table 5, Fig. 4, Fig. 7 and Fig. 5a need to be mentioned in the text before the corresponding images.

(7) Figure 7 is of low quality and needs to be replaced by a high-resolution image with an enlarged scale in the figure.

(8) Line 333, "4. CONCLUSIONS" should be changed to "4. Conclusions".

In summary, the manuscript is recommended for Minor revision.

Author Response

Dear Reviewer,

On behalf of all authors, I would like to thank you for taking the time to read our manuscript and put in your comments, which improved the quality of our work. Below you can find our answers related to each of your comments.

  1. Initial capitalisation in titles should be consistent.

Ad.1. We have corrected the capitalization in the title.

  1. The methodology and purpose of the study in the abstract were not detailed enough.

Ad.2. The abstract has been improved

  1. Line 117, and Line 136, "2.1 ***" was repeated.

Ad.3. We have corrected the numbering of the titles.

  1. Line 212, "3. RESULTS AND DISCUSSION" was revised to "3. Results and discussion".

Ad.4. Line 212, "3. RESULTS AND DISCUSSION" changed to "3. Results and discussion."

  1. Line 289, "Fig. 4. Results of DIC analysis ......", DIC needs to be written as Digital Image Correlation to make the figure title self-explanatory. 

Ad.5.  Line 289, "Fig. 4. Results of DIC analysis ......", DIC we have written as Digital Image Correlation to make the figure title self-explanatory. 

  1. Table 5, Fig. 4, Fig. 7 and Fig. 5a need to be mentioned in the text before the corresponding images.

Ad.6. We added the missing references to the text for tables and figures.

  1. Figure 7 is of low quality and needs to be replaced by a high-resolution image with an enlarged scale in the figure.

Ad.7. We have improved the quality of the photos.

  1. Line 333, "4. CONCLUSIONS" should be changed to "4. Conclusions".

Ad.8. Line 333, "4. CONCLUSIONS" changed to "4. Conclusions".

Reviewer 2 Report

Comments and Suggestions for Authors

Paper entitled "Eco-friendly and innovative approach in building engineering: production of Cement-Glass Composite Bricks with recycled polymeric reinforcement" is a very interesting paper, with many applied innovations.

I suggest accepting it in its present form.

Author Response

Dear Reviewer,

On behalf of all authors, I would like to thank you for taking the time to read our manuscript and put in your acceptance decision. We are honored that you find our work valuable enough to promote it for publication. 

Reviewer 3 Report

Comments and Suggestions for Authors

The article presents a novel approach to producing Cement-Glass Composite Bricks (CGCB) reinforced with 3D-printed recycled PET-G. It showcases the potential for using recycled materials in construction, contributing to environmental sustainability. The study effectively demonstrates improved mechanical and thermal properties of CGCBs, offering a promising alternative to traditional construction materials.

The research is highly relevant to current environmental concerns and showcases innovative use of recycled materials in construction. The experimental procedures and material selection are well-detailed, providing a clear understanding of the research process. The results are well-presented with comprehensive data analysis, supporting the conclusions drawn. This research is a significant contribution to sustainable building materials. It successfully bridges a gap in current knowledge and opens avenues for further exploration. However, few improvements are needed.

1. Table 1: how the composition was obtained? Provide the method for the chemical analysis if it was performed.

2. Table 3: how this proportion was chosen?

3. Section 2.4: It is better to provide a scheme of the setup for DIC measurements to make this part clear for readers.

4. Table 5: add error bars to the values

5. Fig. 7: the values on the color scale are too small, enlarge them

6. The Conclusions section is too long, make more compact: highlight the key findings, reflect on the innovative aspect, place the results within the broader context of sustainable construction and materials science.

Author Response

Dear Reviewer,

On behalf of all authors, I would like to thank you for taking the time to read our manuscript and put in your comments, which improved the quality of our work. Below you can find our answers related to each of your comments.

  1. Table 1: how the composition was obtained? Provide the method for the chemical analysis if it was performed.

Ad.1. We put an additional description just before the mentioned table

  1. Table 3: how this proportion was chosen?

Ad.2.We put an additional description just before the mentioned table

  1. Section 2.4: It is better to provide a scheme of the setup for DIC measurements to make this part clear for readers.

Ad. 3. The mentioned setup is shown in additional figure 3. 

  1. Table 5: add error bars to the values

Ad.4.We have added error bars to the values in Table 5

  1. 7: the values on the color scale are too small, enlarge

Ad.7. We have improved the quality of the whole figure.

  1. The Conclusions section is too long, make more compact: highlight the key findings, reflect on the innovative aspect, and place the results within the broader context of sustainable construction and materials science.

Ad.6. We rephrased our conclusion based on your comment.